# Semi-supervised Ensemble Learning with Weak Supervision for Biomedical Relation Extraction

**Antonios Minas Krasakis**                          AMKRASAKIS@GMAIL.COM
*University of Amsterdam, Science Park 904,*
*1098XH Amsterdam, The Netherlands*

**Evangelos Kanoulas**                                E.KANOULAS@UVA.NL
*University of Amsterdam, Science Park 904,*
*1098XH Amsterdam, The Netherlands*

**George Tsatsaronis**                          G.TSATSARONIS@ELSEVIER.COM
*Elsevier B.V., Radarweg 29,*
*1043NX Amsterdam, The Netherlands*

## Abstract

Natural language understanding research has recently shifted towards complex Machine Learning and Deep Learning algorithms. Such models often outperform their simpler counterparts significantly. However, their performance relies on the availability of large amounts of labeled data, which are rarely available. To tackle this problem, we propose a methodology for extending training datasets to arbitrarily big sizes and training complex, data-hungry models using weak supervision. We apply this methodology on biomedical relation extraction, a task where training datasets are excessively time-consuming and expensive to create, yet has a major impact on downstream applications such as drug discovery. We demonstrate in two small-scale controlled experiments that our method consistently enhances the performance of an LSTM network, with performance improvements comparable to hand-labeled training data. Finally, we discuss the optimal setting for applying weak supervision using this methodology.

## 1. Introduction

The amount of scientific papers in the biomedical field is ever increasing. Published papers contain important information, however, encoded in unstructured text, making it difficult for researchers to locate it. Extracting this information in a structured format and storing it within a knowledge base can have a remarkable impact on a variety of important tasks, ranging from drug design to detecting adverse drug effects. During the past decade, there have been efforts towards automation of Information Extraction [Wei et al., 2016, Krallinger et al., 2017a, Wei et al., 2015], due to the fact that manual annotation of documents from domain experts is labor-intensive to perform on a large scale [Krallinger et al., 2017b].

The broader focus of this work is to help automation of semantic triple extraction from biomedical abstracts. We apply our methodology on two different relations: (a) *Regulations*, indicating that a *Chemical* increases (up-regulates) or decreases (down-regulates) the production of a *Protein* (CPR) and (b) *Chemically Induced Diseases* (CID). Both relations are particularly important for areas such as drug design, safety, and discovery, as it will enable researchers to filter out or select chemical substances with specific properties, faster [Krallinger et al., 2017b, Li et al., 2016].

| Relation | Subject | Object | Dataset |
|---|---|---|---|
| Regulation (CPR) | Chemical | Protein/Gene | BioCreative VI - CHEMPROT |
| Chemically-Induced Disease (CID) | Chemical | Disease | BioCreative V - CDR |

Table 1: Relationships and datasets

Extracting semantic triples is a structured Information Extraction problem. First, one needs to identify the entities of interest (subjects & objects) in the unstructured text, optionally distinguishing them with a unique identifier, and then build a classifier to recognize whether the text describes the target relationship (Relation Extraction).

Our main focus lies on the subtask of relation extraction. As this is a complex and challenging task, increasing the capacity of the learning algorithm is justified, but should coincide with an appropriate increase in the training dataset size [Goodfellow et al., 2016]. However, annotating training datasets for this task is an excessively labor-intensive process. To tackle this problem, we propose a new methodology based on weak supervision, which combines and incorporates ideas from semi-supervised and ensemble learning. The outline of this methodology is the following: (a) We train multiple base learners on a small labeled dataset and use them to predict the labels of a bigger unlabeled dataset; (b) we then use a denoiser to derive weak labels for the unlabeled set using the predictions of the base learners; and (c) we finally train a strong meta-learner using weak supervision on the (previously) unlabeled set.

Our main contributions include: (a) proposing a detailed methodology specific to relation extraction, which can be adapted and generalized to most supervised learning tasks; (b) demonstrating the effectiveness and usability of this methodology in a small-scale controlled experiment; (c) investigating the effect of various denoising methods on the overall system behavior and performance.

To ensure reproducibility of our results and encourage the research community to further experiment with this methodology, we release the code in our GitHub repository[1].

## 2. Related work

In this section we discuss literature related to three topics: information extraction, relation extraction from biomedical text, and semi-supervised and ensemble learning methods.

### 2.1 Information extraction

Information extraction is typically modeled as a fully-, semi- or un-supervised learning problem. Unsupervised methods, such as Open Information Extraction [Banko et al., 2007], do not make use of any training data, but they can only be used for unstructured Information Extraction. On the contrary, fully-supervised methods rely on labeled examples, which have to be manually annotated. Semi-supervised or bootstrapping methods are similar to our approach, as they try to leverage both labeled and unlabeled data. One of the first bootstrapping algorithms was DIRPE [Brin, 1998], which starts from some seed (positive)

---

1. https://github.com/littlewine/snorkel-ml/

examples, extracts patterns from them and use them for finding new positive examples. Other semi-supervised algorithms include Snowball [Agichtein and Gravano, 2000], Know-ItAll [Etzioni et al., 2004] and TextRunner [Banko et al., 2007]. Recent approaches also consider contextual data augmentation, which paraphrases the training samples [Kobayashi, 2018]. All of the aforementioned work focuses on bootstrapping new data to avoid excessive manual annotation, however, different from this work, it does not use the combination of learning algorithms to do so.

Distant supervision [Mintz et al., 2009] is a method to generate weak labels for unlabeled data on relation extraction problems. Specifically, it uses Knowledge Bases (KB) for the creation of weak (positive and negative) labels instead of pre-trained classifiers. Despite the creation of many noisy examples, this approach has proven to be beneficial on large-scale (or web-scale) datasets, as no human annotation is required. The core idea of weakly-labeled datasets remains the same, but is achieved by different means. Our work is complementary to distant supervision, as such algorithms can constitute weak classifiers in our framework.

## 2.2 Relation extraction from biomedical text

Most of the research on biomedical relation extraction has been motivated by BioCreative competitions and their corresponding annotated documents (datasets). BioCreative V (CID task) focused on extracting Chemically-induced Diseases on a document-level [Wei et al., 2015]. The best performing team implemented an ensemble of Support Vector Machines [Xu et al., 2015], while more recent research demonstrated that extending the training set using distant supervision improves the performance [Peng et al., 2016]. A following competition (BioCreative VI - CPR task) focused on identifying relations between Chemicals and Proteins (or Genes) on a sentence level [Krallinger et al., 2017a]. The best performing team implemented an ensemble of LSTM, CNN and SVMs, using majority voting and stacking [Peng et al., 2018]. The second highest score was achieved using a Support-Vector Machines algorithm with a rich set of features, while other approaches using solely Deep Neural Networks demonstrated overfitting problems [Mehryary et al., 2018].

Those results highlight the importance of the lack of training data in this particular domain and demonstrate the suitability of ensemble methods for improving generalization, especially when Deep Neural Networks are used. To that end, our work aims to combine the advantages of those techniques with semi-supervised learning, a subject which to the best of our knowledge has not been investigated for this task. Further, we provide a framework where all of the aforementioned work can be used as an input.

## 2.3 Semi-supervised and ensemble learning methods

Both semi-supervised and ensemble learning techniques aim to improve the performance of Machine Learning models. Ensemble learning typically reduces high variance by combining multiple learners, while semi-supervised learning tries to take advantage of unlabeled data to improve generalization. Although ensembles have been studied thoroughly and used in many applications, their combination with semi-supervised learning has not. Their combination has not been thoroughly studied, despite indications that they can be beneficial to each other [Zhou, 2011]. Ensembles can enhance semi-supervised learning by providing multiple views and therefore allow the system to perform better, earlier (using less data). On the

other hand, expanding the dataset with unlabeled data is likely to increase diversity between the learners. The first system of this kind proposed was co-training [Blum and Mitchell, 1998], a paradigm where two distinct (independent) learning algorithms take advantage of unlabeled data. Following research indicated that complete independence was too luxurious for a real-world setting and the method can even succeed without it, given an expansion on the underlying data distribution [Nigam and Ghani, 2000, Balcan et al., 2005].

Recent work uses expert-defined lexicons and an auxiliary natural language processing system to create noisy annotations, and incorporate co-training to reduce noise and augment signal in distant supervision. Without using any manually labeled data, their system learned to accurately annotate data samples in functional genomics and outperform state-of-the-art supervised methods trained on tens of thousands of annotated examples [Grechkin et al., 2017].

Tri-training [Zhou and Li, 2005] is an extension of co-training to three learners. In this case, when two learners agree on the label of a new data point, they teach (re-train) the third learner using this example. Co-forest [Li and Zhou, 2007] is an extension to even more learners, where the decision on whether an unlabeled example should be added in the re-training stack is made by an ensemble system, using all learners.

The fundamental difference between our methodology and those earlier described, is that the learners included in the ensemble system (base learners) are not used for the final prediction; they are only used as a mean for generating the weak labels. For this reason, we do not aim to re-train or improve the base learners. This allows us to use all of the unlabeled data, whereas in the paradigms above only a few examples annotated with high confidence were added to the stack for re-training.

Last, a different line of research has focused on learning language representations and tuning these representations towards specific machine learning tasks using a small labeled set [Peters et al., 2018, Clark et al., 2018, Devlin et al., 2018]. This work is complementary to our work, and it could comprise our weak classifiers.

## 3. Background on Weak Supervision and Data Programming

In this section, we describe weak supervision and the data programming paradigm, which have both heavily influenced the development of our methodology.

Weak supervision revolves around the idea of training models using labels of questionable quality [Dehghani et al., 2017a,b, Mintz et al., 2009]. Data programming is a paradigm for programmatic creation of such training sets, focusing on the scenario where no ground-truth labels are available [Ratner et al., 2016, 2017b]. It can be outlined in the following steps:

*(1) Provide weak supervision sources:* We define $K$ weak supervision sources and encode them into *Labeling Functions (LF)*. For each unlabeled data point, those sources can either provide a label or abstain from voting. Typically, a LF might consist of a textual pattern, a crowd-worker or a distant supervision resource. However, it is possible to incorporate any other kind of weak source which can provide training labels. We apply the LFs over $M$ unlabeled data points and derive a (possibly incomplete) vote matrix $\Lambda \in \mathbb{R}^{KxM}$.

*(2) Denoising:* Our objective is to derive $M$ weak labels from $\Lambda$ , which are as close as possible to the (unknown) true labels. As a denoiser, data programming uses a probabilistic graphical Generative Model (GM), relying on agreements and disagreements. It

incorporates as trainable parameters: the probability (a) that a LF will label a data point and (b) that this label is correct (accuracy). The structure of the GM is a hyperparameter, representing correlations of the Labeling Functions and can be estimated automatically [Ratner et al., 2016, 2017a, Bach et al., 2018, 2017]. To train the GM without access to the ground truth, data programming maximizes the marginal log-likelihood that the observed votes occur under the Generative Model (for all possible ground-truth labels):

$$\hat{w} = argmax_w log \sum P_w(\Lambda, Y)$$

Further, we use the predicted label distributions $\hat{Y}_i = P_{\hat{w}}(Y_i|\Lambda)$ as probabilistic weak labels.

*(3) Train a noise-aware discriminative model:* We use the generated labels for training and use this model as the final predictor. During training, we minimize a noise-aware variant of the loss function $l$ with respect to the probabilistic weak labels $\hat{Y}$:

$$\hat{\theta} = argmin_\theta \sum_{i=1}^{M} \mathop{\mathbb{E}}_{y \sim \hat{Y}_i} [l(h_\theta(X_i), y)]$$

## 4. Methodology

Based on the concepts of weak supervision and data programming, we propose a methodology for semi-supervised learning, with the intent to capitalize the advantages of multiple learners. In contrast to the scenario where no ground-truth labels are available, we assume that a gold-labeled training set $(D_B)$ is available, but its size is insufficient for training a complex (and therefore more data-hungry) model. We advocate that it would be beneficial to augment additional, lower quality training data to scale the dataset size. Instead of relying on heuristics or crowd-sourced labels, we use machine learning models of lower complexity as weak supervision sources. This comes with the major advantage of adapting an already implemented pipeline with little or no additional effort to similar tasks.

### 4.1 Data collection

In terms of data, we assume the existence of a labeled training set $D_B$ of size $m$, relevant to our task $T$. Additionally to that, we require an unlabeled, arbitrarily large dataset $D_U$ of size $M \gg m$, with the requirement of being drawn from the same distribution as $D_B$. Other requirements are a validation set $D_V$ for hyperparameter tuning and a held-out test set $D_T$ for evaluation purposes.

### 4.2 Constructing diverse base learners

We use $D_B$ to train $K$ base learners on solving $T$. As in a typical ensemble learning scenario, we try to maximize their individual performance while making them capture different "'views"' of the data. To produce multiple learners we rely on varying hyperparameters and design choices throughout the relation extraction pipeline, which results in the creation of 162 base learners. The most important design choices are:

*(1) Sentence pruning:* It is often the case, that words appearing within a sentence might be irrelevant to the entities of interest. For this reason, one can keep only the words between

the two entities of interest or additionally include words appearing within a certain sized window before/after them. More complex approaches incorporating syntactic information can also be considered. One way to do so is to construct the dependency-based parse tree and include only words contained within the Shortest Path connecting the entities of interest. In this work we investigate (a) Whole sentences, (b) window of 0, (c) window of 5, and (d) Shortest Dependency Path (SDP).

*(2) Sequential features:* Additionally to the simple bag-of-words approach, it is also possible to include contiguous sets of tokens as features. In our approach, we use up to tri-grams.

*(3) Text representation:* To convert our corpus to a numerical representation, we use token occurrences (binary counts) or TF-IDF weights.

*(4) Machine learning algorithms:* When the feature matrix is ready, we employ different machine learning algorithms, including Logistic Regression, Support Vector Machines (using Gaussian and linear kernels), Random Forest Classifiers, Long-Short Term Memory Networks and Convolutional Neural Networks. It is important to note, that when the last two models (LSTMs & CNNs) were used, some of the aforementioned feature engineering steps, were not applicable.

### 4.3 Base learner selection

After producing the base learners, we select only a subset of them. This is necessary due to computational cost and the fact that we should avoid including many similar classifiers in a disagreement-based method. Hence, our objective is to maximize both the individual performance of the base learners and their diversity. Since this is complex and still an open issue [Zhou, 2011], we resort to a simple method, where we discard all classifiers having a lower performance than a certain threshold (evaluated on $D_V$), while maximizing diversity. Setting a performance threshold is also desirable, due to the fact that the base learners where automatically created with limited hyperparameter tuning. We set this threshold above the random guess baseline, but low enough to allow less accurate and more diverse classifiers to be part of the ensemble.

To select the most diverse classifiers, we employ a similarity-based clustering method. Using the predictions of the $K$ base learners on $D_V$, we construct a $KxK$ similarity matrix. In this matrix, each row and column refers to a base learner, while each cell consists of the corresponding pairwise inter-annotator agreement rates (Cohen Kappa coefficient). We perform K-means clustering [MacQueen et al., 1967] on this matrix and pick the base learners closest to the cluster centroids as most representative of their cluster. To pick an appropriate number of clusters (and therefore base learners), we refer to the silhouette score coefficient [Rousseeuw, 1987].

### 4.4 Producing weak labels

We predict the labels of $D_U$ using the selected base learners and obtain a $KxM$ (binary) prediction matrix, containing the "knowledge" our base learners have distilled from $D_B$. Consequently, we use a denoiser to reduce the vote matrix into $M$ weak labels. We use the probabilistic *Generative Model* of data programming (described in Section 3) and select the hyperparameters using the validation dataset. Additionally, we consider two simpler

denoisers to unify the label matrix: (a) a *Majority Vote* denoiser producing binary weak labels and (b) an *Average Vote* denoiser calculating an unweighted average of all base learner votes, therefore producing marginal weak labels.

## 4.5 Training a meta-learner

In the last step, we use a discriminative model (as described in Section 3) as a meta-learner. In practice, when training the meta-learner with weak supervision, we trade label quality for quantity. This can be proven to be beneficial in cases where the performance of the meta-learner is upper-bounded by the training set size. By using high-capacity models such as Deep Neural Networks as meta-learners, we allow them to learn their own features and hopefully build a more accurate representation by relying on a much larger, even though noisy, training dataset.

## 5. Experimental Setup

To perform our experiments we use part of the functionality of Snorkel [Ratner et al., 2017b,a], a framework build for relation extraction with data programming and weak supervision.

## 5.1 Datasets

In our experiments, we use the official BioCreative CHEMPROT and CDR datasets [Krallinger et al., 2017a, Li et al., 2016], which consist of annotated PubMed [2] abstracts (split in a training, development and test set – Table 2).

| Dataset | # docs | # candidates |
|---|---|---|
| CHEMPROT (training) | 1020 | 9917 |
| CHEMPROT (development) | 612 | 6227 |
| CHEMPROT (test) | 800 | 8285 |
| CDR (training) | 900 | 8272 |
| CDR (development) | 100 | 888 |
| CDR (test) | 500 | 4620 |

Table 2: Basic dataset statistics

As described in Subsection 4.1, our methodology requires three gold-labeled datasets, along with a held-out test set. In both cases, we use the original test sets as the held-out test set ($D_T$) and report the final scores there. We merge and shuffle the original training and development sets to create the remaining three datasets: We use one part to train the base learners ($D_B$) and another part for validation and hyperparameter selection ($D_V$). We use the remaining documents as if they were unlabelled, $D_U$. The restructured dataset statistics are available in Table 3.

Using this setup, we can make sure that two important requirements are satisfied: (a) there is no bias during the document selection process, i.e. we ensure that the training,

---

2. https://www.ncbi.nlm.nih.gov/pubmed/

validation, test and unlabeled dataset are drawn from the same distribution, and (b) all documents have gone through the same pre-processing steps., i.e. labeled datasets have manually annotated entities, while a randomly drawn unlabeled dataset would require applying some Named Entity Recognition algorithm. We do so to better control the effect of such choices on the results of our algorithm. Another advantage with this controlled approach is that we are able to compare the performance of the meta-learner trained with weak supervision to the optimal performance, which would be achieved if the ground-truth labels were available.

## 5.2 Text pre-processing and named entity recognition

Most of the steps in our text pre-processing pipeline are performed by SpaCy (v1.0), an open-source library for Natural Language Processing. More specifically, SpaCy performs the following tasks in our pipeline: Sentence splitting, Tokenization, and Dependency parsing. Both datasets contain manually annotated Named Entity Tags, which are required for the Candidate Extraction step (Subsection 5.3).

## 5.3 Candidate extraction

Given all entities of interest within the text, we look for relationship candidates. We do not consider cross-sentence relations and proceed only with candidates found within the same sentence. We use Snorkel [Ratner et al., 2017b] for candidate extraction and mapping the candidates to their ground-truth labels.

## 5.4 Entity replacement

A relationship clasifier is important to understand the Natural Language, rather than memorize the pairs which interact with each other. For this reason, we replace all entities of interest using the tokens 'ENTITY1' and 'ENTITY2' for the entities we want to predict. We also replace additional entities of the same type within the same sentence with the tokens 'CHEMICAL', 'GENE' or 'DISEASE' accordingly.

## 5.5 Meta-learner selection & training

In our experiments, we use a simple bi-directional Long-Short Term Memory network, which is one of the most commonly used and better performing Deep Neural Networks, on tasks related to Natural Language. We use randomly initialized word embeddings and perform random under-sampling to keep an equal class balance. We also try different hyperparameter settings, including different dropout values $(0, 0.25$ and $0.5)$ and training epochs (1-30). We make the selection based the validation dataset $D_V$.

## 6. Research Questions and Experimental Design

In this section, we form the research questions that we aim to answer, accompanied by our motivation to explore them. We discuss more details regarding RQ1 and RQ2 in Subsections 6.1 and 6.2, respectively.

RQ1 Can we enhance biomedical relation extraction when using Machine Learning classifiers as sources of weak supervision?

RQ2 Which is the optimal setting for using weak supervision on this task?

## 6.1 Machine learning classifiers as weak supervision sources

Related literature provides theoretical warranties, that in this setting and given specific requirements, adding weakly labeled data will improve the performance of the meta-learner [Ratner et al., 2016]. Additionally, as the amount of weakly labeled data increases, the performance of the meta-learner is expected to improve quasi-linearly (almost as good) compared to the scenario where the ground-truth labels were provided [Ratner et al., 2017a]. Those requirements include that the weak supervision sources should have accuracy better than random guess, overlap and disagree with each other enough (so that their accuracy can be estimated) while capturing different 'views' of the problem (diversity). In other words, when combined they should be able to model the problem space sufficiently, so that meaningful weak labels can be produced.

However, to the best of our knowledge, Machine Learning classifiers have not been used so far as weak supervision sources in such a setting, nor in this specific task. Therefore, it is unclear whether there is a diverse and sufficiently big set of base learners, that satisfies those conditions after being trained on the same dataset. This is a critical question, which actively affects the usability of the described methodology. To evaluate whether weak supervision helps, we conduct several experiments under different setups and compare the performance of the meta-learner when trained on three different modes: (a) full-supervision on $D_B$, (b) weak-supervision on $D_U$, (c) weak-supervision on $D_U$ combined with full-supervision on $D_B$. Additionally, we evaluate whether weak-supervision can achieve results comparable to full-supervision, after training the meta-learner using all ground-truth labels ($D_U + D_B$).

## 6.2 Optimal setting for applying this methodology

**Number of base learners:** Selecting the optimal number of classifiers to be used as base learners is not a straightforward task. Naturally, we can construct only a few top-performing learners and as we add more of them, we start to sacrifice performance in favor of diversity [Zhou, 2011]. To examine this, we gradually increase the number of Base Learners while benchmarking the performance of the weak labels and the meta-learner.

**Comparison of various denoising methods:** The denoising component is fundamental to this method, as it dictates the quality of the weak labels the final learner will be trained on. We use the three denosing methods described in Subsection 4.4 and assess the quality of the results.

**Meta-learner performance under different weak label distributions:** The denoiser can produce either binary or marginal (non-binary) weak labels. Additionally, marginal weak labels might be following different distributions, ranging from extremely U-shaped (almost binary) to more uniformly distributed. We perform an error analysis to investigate their effect on the training and the final performance of the meta-learner.

## 7. Results and Analysis

In Subsection 7.1 we answer RQ1, that is whether supervised machine learning classifiers can be used as weak classifiers, while in Subsection 7.2 we answer RQ2, that is what is the optimal setting for applying weak supervision.

### 7.1 Weak classifiers

To select the base learners, we use the strategy described on Subsection 4.3. We experiment with different number of base learners and benchmark results in intervals of 5 and where the silhouette scores are maximized (Table 4). Specifically, we report performance of (a) the Base Learners (in detail in Appendix A), (b) the weak labels/marginals produced by the denoisers and (c) the meta-learner when trained on those weak labels.

| task | training subset | #docs | training datapoints | training datapoints (undersampled) | F1 score |
|---|---|---|---|---|---|
| CPR | $D_B$ | 400 | 3840 | 2060 | 44.7% |
| | $D_U$ | 926 | 8221 | 4516 | 54.0% |
| | $D_B + D_U$ | 1326 | 12061 | 6576 | 55.3% |
| CID | $D_B$ | 300 | 2860 | 2156 | 49.5% |
| | $D_U$ | 600 | 5576 | 4166 | 55.1% |
| | $D_B + D_U$ | 900 | 8436 | 6322 | 57.4% |

Table 3: LSTM performance with gold labels

| task | # | min | mean | max | Majority & AVG Vote | Generative Model | using M.V. labels | using AVG marginals | using GM marginals |
|---|---|---|---|---|---|---|---|---|---|
| | | | performance (F1) | | F1 score of weak labels | | F1 score of meta-learner (LSTM) | | |
| CPR | 5 | 53.33 | 54.15 | 55.73 | 58.91 | 61.07 | 50.11 | 53.56 | 52.40 |
| | 10 | 44.43 | 53.94 | 60.8 | 63.04 | 62.22 | 52.66 | 56.45 | 52.93 |
| | 13 | 44.43 | 53.48 | 60.8 | 63.09 | 61.28 | 54.16 | 58.03 | 52.65 |
| | 15 | 44.43 | 53.16 | 61.3 | 63.11 | 61.48 | 51.01 | 56.32 | 53.40 |
| | 20 | 44.43 | 53.39 | 61.3 | 63.57 | 61.91 | 50.05 | 55.60 | 54.73 |
| CID | 5 | 49.70 | 51.85 | 56.21 | 53.36 | 56.31 | 54.29 | 53.68 | 52.27 |
| | 6 | 49.70 | 51.60 | 56.21 | 55.95 | 54.15 | 52.14 | 53.48 | 54.23 |
| | 10 | 49.85 | 51.44 | 56.21 | 56.90 | 54.09 | 53.28 | 54.80 | 54.61 |
| | 13 | 49.85 | 51.21 | 56.21 | 56.71 | 55.37 | 53.41 | 54.75 | 53.84 |
| | 15 | 49.84 | 51.17 | 56.21 | 57.21 | 56.18 | 53.91 | 55.23 | 54.69 |
| | 20 | 49.70 | 51.60 | 56.95 | 57.30 | 56.71 | 54.86 | 55.44 | 55.68 |

Table 4: Performance achieved with weak supervision

We compare the results achieved with weak supervision on $D_U$ (Table 4), versus full-supervision (Table 3). It is evident that training the meta-learner with weak labels ($D_U$)

and a $2 - 2.5x$ increase in the training set size always performs better compared to training with the fewer, gold labels ($D_B$). Performance is further improved, when we also include the available ground-truth labels ($D_B + D_U$). This proves that we can successfully augment additional training data using weak supervision, as long as they are drawn from the same distribution.

Additionally, weak supervision can often achieve a performance comparable to full-supervision; in other words almost as good as if we were using the ground-truth labels. Interestingly enough, there are even cases where weak supervision achieves slightly better results. However, those differences are minor and not statistically significant, due to high variance on the meta-learners' performance. We also recognize that whenever this happened, the under-sampled training set size of the final learner in weak supervision was bigger, due to the fact that undersampling was based on the weak labels instead of the real ones.

Another noteworthy observation, is that a simple Majority Vote often outperforms the meta-learner. However, this is an expected result and does not undermine the importance of our results, as we can verify from Table 3 that this model (LSTM) cannot outperform Majority Voting with such a small training dataset, even when gold quality labels are used.

At last, we visualize the learning curves of the meta-learner (starting from the ground-truth labels) to ensure that the weak labels are meaningful and do actually improve performance. Figures 1b, 1c and 1d indicate an upward trend, while the outlined confidence intervals indicate that the results are statistically significant. Moreover, we observe that the F1 score on the training set is always much higher than the test score. This is a sign that our meta-learner suffers from high variance (overfitting), as the model capacity is far from being fulfilled. Therefore, additional training data are expected to improve the performance of the meta-learner.

## 7.2 Optimal setting

The complexity of the problem and the methodology, along with the small dataset size do not allow us to draw definite answers in some of the following questions. However, we will perform an analysis based on our experimental results and discuss our findings.

**Number of base learners:** We can see from Table 4 that the $F1$ score of the weak Majority Vote labels for 5 learners is the lowest in both experiments. When it comes to the Generative model weak marginals, we cannot observe any significant pattern, as the F1 score always deviates within 1.5 & 2.5 points respectively. The performance of the meta-learner when trained with Average Vote marginals deviates to a certain extent when more than 10 base learners are used, but always performs better compared to when only 5 base learners are used. Using Generative model marginals, performance seems to slightly improve as the number of base learners increases, with two exceptions.

**Comparison of various denoising methods:** In the vast majority of cases, the meta-learner achieves the best performance when trained with Average Marginals. Generative Model marginals also seem to improve its performance compared to Majority Vote weak labels, with one exception. However, it is worth highlighting that GM marginals depend on hyperparameters, which are chosen based on the F1 score on a validation dataset. Later on this section, we argue and demonstrate why this particular measure cannot fully reflect the

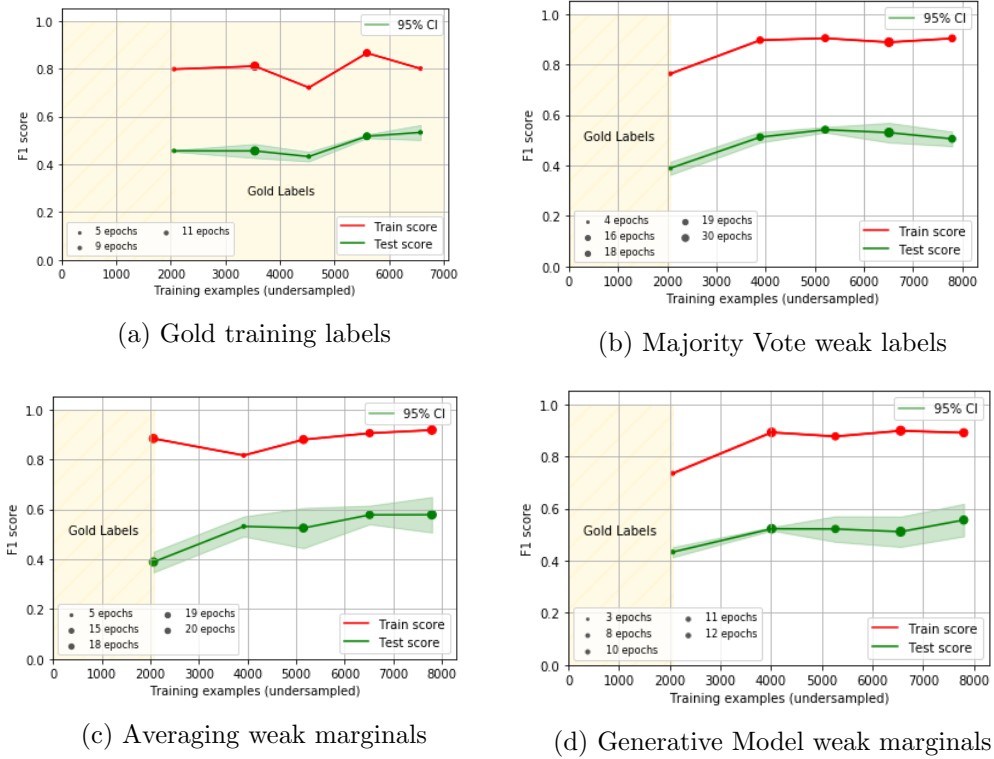

(a) Gold training labels

(b) Majority Vote weak labels

(c) Averaging weak marginals

(d) Generative Model weak marginals

Figure 1: LSTM learning curves

quality of marginal weak labels. Therefore, it is not certain whether we have achieved the optimal performance in cases where the Generative Model was used as the denoiser.

**Performance under different weak label distributions:** The denoisers can produce weak labels which are either binary or marginal (non-binary). We can conclude that marginal weak labels improve the performance of the meta-learner compared to binary labels. This is a straightforward comparison, as Majority Vote weak labels always perform worse than Average marginals while none of them is subject to hyperparameter tuning.

Moreover, we observe that the Generative Model tends to create marginals following a U-shaped distribution (close to 0 or 1) in contrast to the average marginals, which are spread more uniformly. This is evident from the error analysis we perform on the validation set, using a classification boundary of 0.5 (Figures 2a and 2b). In both cases, the amount of misclassified weak labels and therefore their F1 score is the same. However, it is evident that Average Vote labels are of higher quality, as most of their misclassified labels are relatively closer to 0.5. This is inevitable, as the vast majority of the GM marginals are very close to 0 and 1. Furthermore, these figures demonstrate the unsuitability of the $F1$ score for the evaluation of marginal weak labels.

Figure 3 shows how the training loss and validation scores change as we train the LSTM for more epochs. We can see that when marginal labels are used, the training error remains relatively high. This is especially true with the Average weak marginals, which are spread

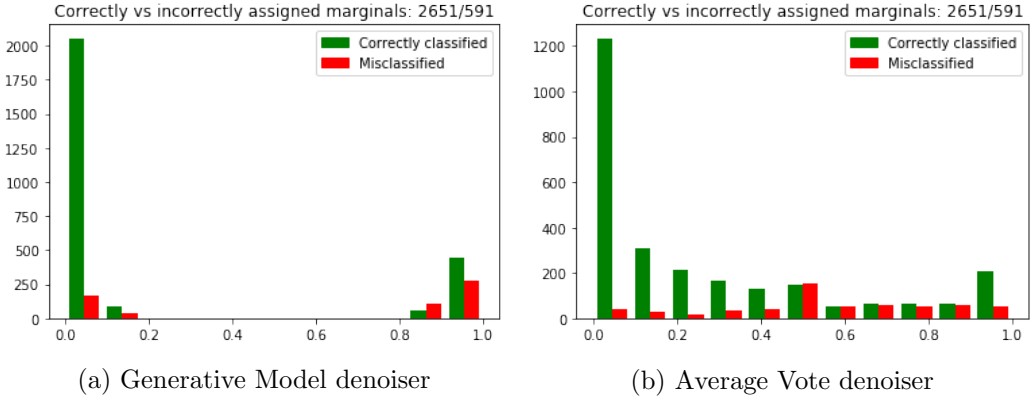

(a) Generative Model denoiser        (b) Average Vote denoiser

Figure 2: Error analysis

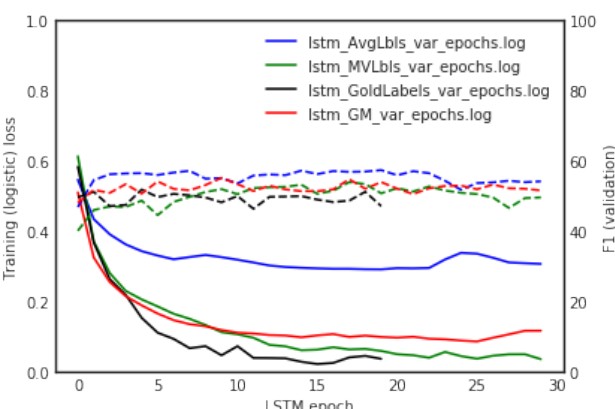

Figure 3: LSTM training loss and validation score per training epoch

more uniformly. On the contrary, it only takes a few epochs for the LSTM to start predicting the binary training labels accurately, despite a small delay on the noisy-labeled MV weak labels. Ultimately, training a classifier using marginal labels can be thought of as a regression problem. In practice, we ask the model to predict an exact number (the output of the denoiser) and penalize it every time it fails to do so.

Lastly, we can see that the distributions of predicted logits (on unseen examples) become more spread as the training marginals distributions become more uniform (Figures 4a and 4b). This is something we would also expect to see when doing regression instead of classification.

## 8. Unlabeled dataset expansion

In this section we discuss our efforts to apply our methodology on the CPR task, while expanding the labeled and unlabeled datasets. We use all of the CHEMPROT documents (excluding validation and test sets) to train the base learners. To construct $D_U$, we use

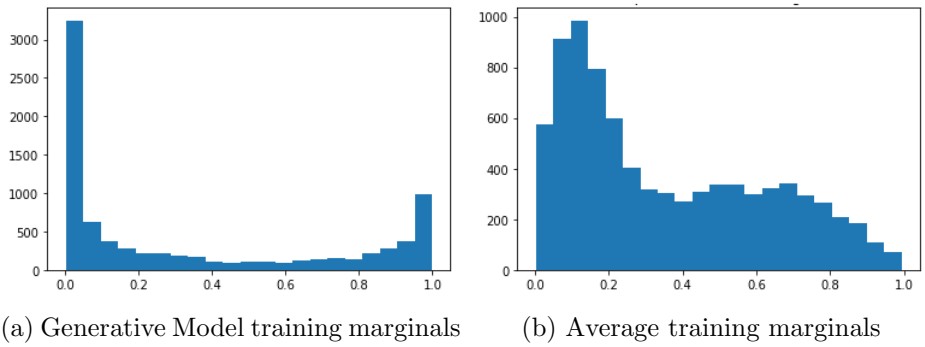

(a) Generative Model training marginals     (b) Average training marginals

Figure 4: Histogram of predicted logits of LSTM

the PubMed API [3] and download two different collections: (a) the outgoing citations of CHEMPROT documents (b) the 25 most similar documents of each CHEMPROT document (using the PMRA similarity metric [Lin and Wilbur, 2007]). In both cases, the performance of the meta-learner decreases as we add weakly labeled data, indicating a problem with the quality of $D_U$ or the generated weak labels. We also observe a predicted class imbalance of 1:14 on the outgoing citations dataset compared to 1:4 on the original, indicating inherently different dataset distributions. To validate this, we use the t-SNE algorithm [Maaten and Hinton, 2008] along with features extracted from our best-performing base-learner and visualize candidate samples drawn only from the original set (Figure 5a) versus samples drawn from both sets (Figure 5b). It becomes evident that most candidates of the new dataset lie in specific regions of the $2D$ space, confirming that it is unsuitable for our use case. To that end, the best practices of constructing appropriate unlabeled datasets, must be further investigated.

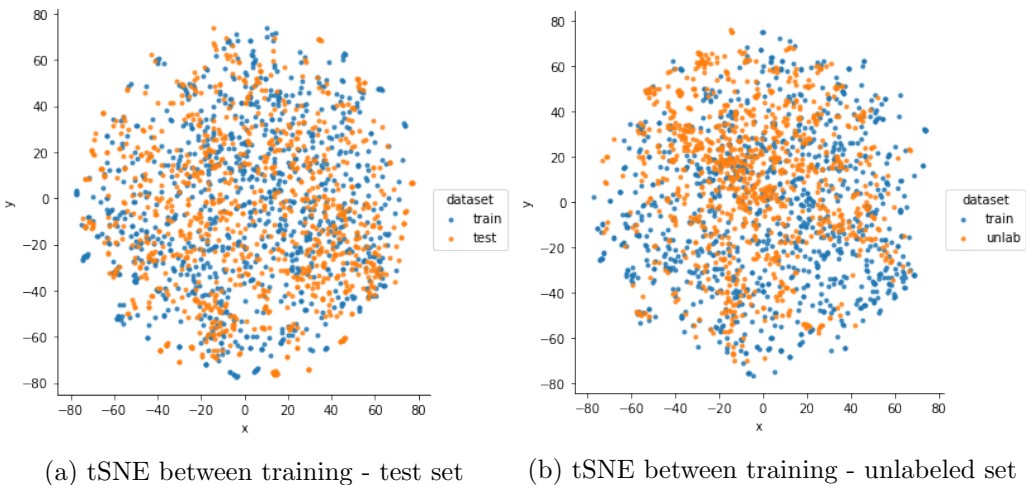

(a) tSNE between training - test set     (b) tSNE between training - unlabeled set

Figure 5: tSNE visualization of data

---

3. https://www.ncbi.nlm.nih.gov/pmc/tools/developers/

## 9. Conclusions & future work

We have shown that weak supervision is a tool which can be used for enhancing the performance of complex models, such as deep neural networks, while utilizing both unlabeled data and multiple base learners. Additionally, we have shown that the proposed methodology is practically feasible for the task at hand, as we have succeeded on defining a combination of base learners, which model the problem space sufficiently and allow us to take advantage of additional, unlabeled data. This comes under the requirement that the unlabeled data are drawn from the same domain/distribution as our labeled data, so that our base learners can generalize and perform adequately on $D_U$.

In practice, our methodology shifts the human effort from hand-labeling examples to feature engineering and construction of diverse learners. More importantly, once a satisfactory set of diverse learners is available, we can use this method to scale the training datasets in arbitrarily high levels while consistently improving the performance over the supervised learning paradigm. Moreover, the same pipeline can be re-used on similar tasks with the only requirement of providing the appropriate datasets. On the contrary, in the typical supervised learning paradigm, we would have to repeatedly hand-label large datasets.

Despite demonstrating the usability of our method using a controlled, small-scale dataset, it is crucial to further explore the requirements of constructing a large enough unlabelled dataset and perform the same experiments there. That would likely improve the meta-learner performance further (which is currently upper-bounded by the small dataset size) and allow us to draw stronger conclusions on the research questions of Subsection 6.2. Additionally it would allow us to inspect how performance improves with the increase of $D_U$ in a different scale of magnitude and if there seems to be a certain performance threshold, which we cannot surpass using weak supervision. Our preliminary experiments demonstrate that collecting an appropriate unlabeled dataset given a labeled one is a challenging task itself, along with the definition of "appropriate", and semi-supervised algorithms should not take the existence of an appropriate unlabeled dataset for granted.

Further, it would be very important to conclude on a more appropriate metric than the F1 score for the evaluation of marginal weak labels. Currently, the absence of an appropriate metric prevents us from drawing conclusions directly from the weak labels, without having to introduce an additional step (train the meta-learner). This would also allow us to select the optimal hyperparameters of the Generative Model and could have a significant impact upon the final performance.

Other areas for further investigation include experimenting with the meta-learner (eg. using pre-trained word embeddings or other model architectures) and defining a more appropriate selection method for the Base Learners. Last, it would be interesting to examine how this system would behave if the Base Learners abstained from voting on the examples they are less certain about. One could simply delete a percentage of the votes which are closer to the classification boundary, or perform a probability calibration on the output of the Base Learners and set a minimum confidence threshold below which they would abstain voting. This could also provide the Generative Model with a modeling advantage, compared to unweighted methods (such as Majority Voting), as described in an analysis related to the trade-offs of weak supervision [Ratner et al., 2017a].

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
