# OpenReview forum: "Semi-supervised Ensemble Learning with Weak Supervision for Biomedical Relationship Extraction"
_AKBC.ws/2019/Conference — AKBC 2019_

### Official Review · AnonReviewer2 · 2019-01-06

**Rating:** 7
**Confidence:** 4

**Review:**

-summary
This paper addresses the problem of generating training data for biological relation extraction, looking specifically at the biocreative chemprot dataset. The authors weakly label data by using a set of weak classifiers, then use those predictions as additional training data for a meta learning algorithm.

-pros
- nicely written paper with good exposition on related works
- good analysis experiments

-cons
- experiments all on a single dataset
- methodology isn’t very novel

-questions
- In table 3, majority vote of weak classifiers outperforms the meta learning. Are these two numbers comparable and if so, does this mean an ensemble of weak classifiers is actually the best performing method in your experiments.

---

> ### Author Response · Authors · 2019-02-05
> **Review response & revision details**
>
> The authors would like to thank the reviewer for his comments.
> Responding to the reviewers request, we repeated the experiments using the very same process/methodology in a different dataset investigating a similar relationship, relevant to the biomedical domain. Experimental results demonstrate an identical behavior, and therefore all of our initial conclusions/remarks still hold. The paper has been revised, containing all of the additional results and details.
>
> Regarding the question about the ensemble of weak classifiers vs meta-learner, it is indeed true that the ensemble outperforms our method for this problem and in a training set of such size. However, this is does not underestimate the importance of our method/results, as the ensemble can not improve using additional unlabelled data, in contrast to our methodology.

---

### Official Review · AnonReviewer1 · 2019-01-09
**The paper proposes a potentially interesting direction, though needs to improve in many aspects.**

**Rating:** 5
**Confidence:** 5

**Review:**

The authors propose to semi-supervised method for relation classification, which trains multiple base learners using a small labeled dataset, and applies an ensemble of them to annotate unlabeled examples for semi-supervised learning. They conducted experiments on a BioCreative shared task for detecting chemical-protein interactions from biomedical abstracts. The experimental results suggest that ensemble could help denoise the self supervised labels. Overall, this is an interesting direction to explore, but the paper can also improve significantly in several aspects.

First, the discussion about related work is a bit lacking and at place slightly misguided. The authors seem to be unaware of a growing body of recent work in biomedical machine reading using indirect supervision, e.g., Deep Probabilistic Logic (EMNLP-18). These methods have demonstrated success in extracting biomedical relations across sentences and beyond abstracts, using zero labeled examples. Likewise, in discussing semi-supervised approaches and cross-view learning, the authors should discuss their obvious connections to recent progress such as Semi-Supervised Sequential Modeling with Cross-View Learning (EMNLP-18) and EZLearn (IJCAI-18).

Additionally, the authors seem to pitch the proposed method against standard weak supervision approaches. E.g., the paper stated "For the creation of the weak labels, we use classifiers pretrained in a small labeled dataset, instead of large Knowledge Bases which might be unavailable." Effectively, the authors implied that distant supervision is not applicable here because it requires "large knowledge bases". In fact, the precise attraction of distant supervision is that knowledge bases are generally available for the relations of value, even though their coverage is sparse and not up-to-date. And past work has shown success with small KBs using distant supervision, such as recent work in precision oncology:
	- Distant Supervision for Relation Extraction beyond the Sentence Boundary (EACL-17)
	- Cross-Sentence N-ary Relation Extraction with Graph LSTMs (TACL-17)

A labeled dataset contains strictly more information than the annotated relations, and arguably is harder to obtain than the corresponding knowledge base. Some prior work even simulated distant supervision scenarios from labeled datasets, e.g., DISTANT SUPERVISION FOR CANCER PATHWAY EXTRACTION FROM TEXT (PSB-15). These won't detract significance in exploring ensemble learning, but the proposed direction should be considered as complementary rather than competing with standard weak supervision approaches.

The paper should also discuss related datasets and resources other than BioCreative. E.g., BioNLP Shared Task on event extraction (Kim et al. 2009) is influential in biomedical relation extraction, followed by a number of shared tasks in the same space.

Another major area that can be improved is the technical and experimental details. E.g.:

The ensemble of base learners is key to the proposed method. But it's unclear from the paper how many base learners have been considered, what are their types, etc. At the high level, the paper should specify the total number of candidate base learners, distribution over major types (SVM, NN, ...), the types of chosen centroids, etc. For each type, the paper should include details about the variation, perhaps in supplement. E.g., SVM kernels, NN architectures.

Table 3 shows the mean F1 score of base learners, which raises a number of questions. E.g., what're the top and bottom performers and their scores? If the mean is 53, that means some base learner learning from partial training set performs similar to LSTM learning from the whole training set, so it seems that out right some learner (supposedly not LSTM) is more suited for this domain? It's unclear whether LSTM is included in the base learner. If it was, then its low performance would mean that some top performers are even higher, so the ensemble gain could simply stem from their superiority. In any case, the lack of details makes it very hard to assess what's really going on in the experiment.

A key point in the paper is that high-capacity methods such as LSTM suffers from small dataset. While in general this could be true, the LSTM in use seems to be severely handicapped. For one, the paper suggested that they undersampled in training LSTM, which means that LSTM was actually trained using less data compared to others. They did this for imbalance, but it is probably unnecessary. LSTMs are generally less sensitive to label imbalance. And if one really has to correct for that, reweighting instances is a better option given the concern about small training set to begin with.

The paper also didn't mention how the word embedding was initialized in LSTM. If they're randomly initialized, that's another obvious handicap the LSTM suffers from. There are publicly available PubMed word2vec embedding, not to more recent approaches like Elmo/Bert.

Minor comments:

In Table 3, what does it mean by "F1 of weak labels"? Does it mean that at test time, one compute the ensemble of predictions from base learners?

The paper uses D_G in one place and D_B in another. Might be useful to be consistent.

"Text trimming" sounds quite odd. What the paragraph describes is just standard feature engineering (i.e., taking n-grams from whole text vs. in between two entities).

Fig 3: it's unclear which line is which from the graph/caption. One has to guess from the analysis,

Fig 5: what was used in tSNE? I.e., how is each instance represented? It's also hard to make any conclusion from the graph, as the authors' tried to make the point about distributions.

---

> ### Author Response · Authors · 2019-02-05
> **Review response & revision details (1)**
>
> The authors would like to thank the reviewer for his thorough and many suggestions.
> Responding to the reviewers request, we repeated the experiments using the very same process/methodology in a different dataset investigating a similar relationship, relevant to the biomedical domain. Experimental results demonstrate an identical behavior, and therefore all of our initial conclusions/remarks still hold. The paper has been revised, containing all of the additional results and details.
> We further describe our changes and respond to this review in our follow-up comment (see below).
>
> Please also note that the authors are planning to release the full source code after the conference, to ensure reproducibility of the results and help the research community adapt the experiment to their own use cases.

---

> > ### Author Response · Authors · 2019-02-05
> > **Review response & revision details (2)**
> >
> > The revision also addresses several other aspects of the review, such as: (a) augmenting the related literature with discussion of the relevant indicated articles, (b) providing thorough details regarding the architecture and performance of the selected base-learners (see Appendix) (c) Addressing specific details indicated by the reviewer.
> > In addition to those changes, we also have some follow-up remarks/responses on this review:
> >
> > (a) Distant supervision:
> > It is indeed true that we could complement our method using distant supervision. However, it is ambiguous what is an “important” relationship and what not, therefore the construction or augmentation of a relationship in a Knowledge base could be much more time-consuming compared to labelling some documents (eg. 400 abstracts) for this relation. In any case, our work is complementary to distant supervision work, which we have now highlight in the revised version.
> > (b) Tackling class imbalance on the LSTM:
> > The reviewer concluded that the LSTM was trained with less examples compared to the other simpler models (Base Learners) and therefore had an inferior performance because of that. This is not true. In all cases, we performed undersampling to address class imbalance. Using the exact same undersampled ground-truth training dataset (D_B), the LSTM performed consistently worse than simpler models.
> > The only training set size mismatch occured when comparing the LSTM (as a meta-learner), using different weak labels (on D_U). This stems from the fact, that the denoiser produces different weak labels in each case. From this point onwards, we simulate a real-world scenario, where we don’t have the real labels of D_U and undersample the meta-learner using the weak (virtually unknown) labels. If we consider case A, where the denoiser produces equally balanced classes on the weak labels, and case B where we have a (predicted) imbalance of 55/45, case B will result in a 10% smaller training set. This does happen to some extent when we compare the denoising methods to each other, and our methodology, compared to the scenario where we would have all gold labels available (for D_B and D_U). However, this detail does not affect by any means the most important comparison; training the LSTM @ D_B with gold labels vs. training the LSTM with weak labels @ D_U.
> > Solving this issue with re-weighting could possibly prove to be a better solution, but the solution is non-trivial and could introduce additional bias to the results. The reason is that this is not the typical classification scenario: the meta-learner is not trained using True/False labels, but with numbers in the range of [0,1] which correspond to confidences. Further, already observed that when gold-labels where used, the LSTM performed better with undersampling vs. changing the class weights in the loss function. Therefore, proceeding with such a design choice would be non-trivial and require further investigation.
> > (c) LSTM suitability as meta-learner:
> > It is already proven by the experiments that the LSTM itself performs lower than the average Base Learner (eg. a simple Logistic Regression), possibly due to the combination of model complexity and few training data.
> >
> > This is a common phenomenon when the model complexity is not on par with the amount of training set size. However, this is beneficial for the nature of our experiments, as we expect to see a much bigger performance gain using a complex model (LSTM) compared to a simple model (Logistic Regression with bag-of words) when increasing the training set size. To put this into numbers, the LSTM had a performance gain of approximately 7-10 F1 points when doubling the training set size, whereas simpler algorithms gained only a few (in principle <5) F1 points. Therefore, in case where a Logistic Regression was to be used as a meta-learner, our experiments would become weaker, as it would become harder to notice performance gains and draw conclusions. Further, the word embeddings were indeed randomly initialized; we now highlight this detail in the revised version.
> >
> > (d) F1 of weak labels:
> > The output of the denoiser is a probabilistic/marginal (or binary) weak label. It ranges from 0-1 and corresponds our confidence that the true label is “True” or False, ie. a label of 0.95 would mean 95% confidence of the true label being “True”. If we draw the classification boundary on 0.5, we can compute the F1 scores of the weak labels themselves (which are later used to train the meta-learner). Several adjustments were made to the paper to further clarify this point.

---

### Official Review · AnonReviewer3 · 2019-01-10
**An interesting paper in methodology, but I'm not entirely convinced by the experiments**

**Rating:** 5
**Confidence:** 3

**Review:**

This paper proposes an interesting combination of semi-supervised learning and ensemble learning for information extraction, with experiments conducted on a biomedical relation extraction task. The proposed method is novel (to certain degree) and intuitively plausible, but I'm not entirely convinced of its effectiveness.

Pros:

- Novel combination of semi-supervised learning and ensemble learning for information extraction
- Good discussion of literature
- Paper is well written and easy to follow

Cons:

- Experimental design and results are not supportive enough of the effectiveness of the proposed method.

Specifically, I think the following problems undermine the claims:

- The experiments are conducted on a single dataset. I understand it may be hard to get more biomedical RE datasets, but the proposed method is in principle not limited to biomedical RE (even though the authors limited their claim to biomedical RE). In general I think it needs multiple datasets to test a generic learning methodology.

- Only one split of training data is used, and the unlabeled data to labeled data are close in scale. It would be really informative if there are experiments and performance curves with different labeled subsets of different sizes. Also, (after under-sampling) 4500 unlabeled vs. 2000 labeled seems less impressive and may be not very supportive of the usefulness of the proposed method. What will happen if there are only 200 labeled examples? 400?

- No comparison with other semi-supervised learning methods for information extraction, so it's not clear how competent the proposed method is compared with other alternatives.

- The fact that the mean base learner performance is often on par with the meta-learner, and simple majority vote of the weak labels can largely outperform the meta-learner may suggest a better meta-learner other than LSTM should have been used.

---

> ### Author Response · Authors · 2019-02-05
> **Review response & revision details**
>
> The authors would like to thank the reviewer for his comments.
> Responding to the reviewers request, we repeated the experiments using the very same process/methodology in a different dataset investigating a similar relationship, relevant to the biomedical domain. Experimental results demonstrate an identical behavior, and therefore all of our initial conclusions/remarks still hold. The paper has been revised, containing all of the additional results and details.
>
> The reviewer also proposed to change the dataset proportions; decreasing the labelled set to increase the “virtually” unlabelled examples to conclude if the overall performance gain of the meta-learner would increase further. While this is an interesting direction, it moves towards few-shot learning, which is a different use case and out of the scope of this paper. This approach comes with additional challenges, mainly related to defining enough adequate base learners which can provide meaningful results in a handful of labelled examples. It is even further more challenging for our selected meta-learner (LSTM), which has shown a big deficiency when trained with only few training labels.
> The performance of the LSTM indeed seems to be significantly lower compared to the rest of the models. It is already proven by the experiments that the LSTM itself performs lower than the average Base Learner (eg. a simple Logistic Regression). possibly due to the combination of model complexity and few training data. This is a common phenomenon when the model complexity is not on par with the amount of training set size. However, using a complex model (LSTM) compared to a simple model (Logistic Regression with bag-of words) is beneficial for the nature of our experiments, as we expect to see a much bigger performance gain when increasing the training set size. To put this into numbers, the LSTM had a performance gain of approximately 7-10 F1 points when doubling the training set size, whereas simpler algorithms gained only a few (in principle <5) F1 points. Therefore, a Logistic Regression, would not be able to make good use of larger dataset, hence the choice of LSTM.
>
> Please also note that we are planning to release the full source code after the conference, to ensure reproducibility of the results and help the research community adapt the experiment to their own use cases.

---

### Meta-Review · Area_Chair1 · 2019-02-12
**Interesting model that makes use of semi-supervised data and meta-learning on top of that**

**Recommendation:** Accept (Poster)
**Confidence:** 4

**Metareview:**

Given a few base-learners, can we learn a meta-learner that performs better than the base-learners? The paper addresses this question in the context of bio-medical relation extraction where training data is often small. A number of base-learners are learned (SVM, LSTM, Logistic Regression etc) on a small training data which are further used to annotate a large amount of unsupervised data. Using data-programming techniques of Ratner et al. 2016, a set of n-weak labels are created. Then a discriminative model (which is called meta-learner here) is further trained on these weak labels to predict the final label.

The reviewers had many concerns on the initial version. Some of the main ones are below:

1) Reviewer 3 suggests using an additional dataset.
2) Reviewer 1 did a great job in suggesting plenty of literature.
3) Reviewer 1 also suggested making several empirical results clear.

The revised version addresses these concerns with several additions. I am satisfied with the revisions, and I believe the paper will be a good addition to the conference. Based on the revised version, I recommend accepting the paper.

A suggestion for improvement: The paper relies a lot on the data programming work of Ratner et al. 2016, 2017, which is explained at a high level in Section 3 and 4. A paper should be self-sufficient when it comes to understanding. Please describe these methods formally and in necessary detail.

---

### Decision · Program_Chairs · 2019-02-15
**AKBC 2019 Conference Decision**

Accept